# Resting state functional connectivity in the human spinal cord

Robert L Barry[1,2]*, Seth A Smith[1,2,3], Adrienne N Dula[1,2], John C Gore[1,2,3]

[1]Vanderbilt University Institute of Imaging Science, Nashville, United States; [2]Department of Radiology and Radiological Sciences, Vanderbilt University Medical Center, Nashville, United States; [3]Department of Biomedical Engineering, Vanderbilt University, Nashville, United States

**Abstract** Functional magnetic resonance imaging using blood oxygenation level dependent (BOLD) contrast is well established as one of the most powerful methods for mapping human brain function. Numerous studies have measured how low-frequency BOLD signal fluctuations from the brain are correlated between voxels in a resting state, and have exploited these signals to infer functional connectivity within specific neural circuits. However, to date there have been no previous substantiated reports of resting state correlations in the spinal cord. In a cohort of healthy volunteers, we observed robust functional connectivity between left and right ventral (motor) horns, and between left and right dorsal (sensory) horns. Our results demonstrate that low-frequency BOLD fluctuations are inherent in the spinal cord as well as the brain, and by analogy to cortical circuits, we hypothesize that these correlations may offer insight into the execution and maintenance of sensory and motor functions both locally and within the cerebrum.

## Introduction

Since the early 1990s, thousands of functional magnetic resonance imaging (fMRI) studies have offered new insights into the functional architecture of the brain and have significantly increased our understanding of normal and aberrant brain functions. The earliest papers investigated task-based fMRI, where evoked BOLD signal changes were interpreted as hemodynamic responses subsequent to neural activity (*Ogawa et al., 1990*; *Bandettini et al., 1992*; *Kwong et al., 1992*; *Ogawa et al., 1992*) and were used to infer which brain regions were activated for a specific task. The range and impact of fMRI methods were expanded in 1995 when Biswal et al. established the existence of correlations between low-frequency (< 0.08 Hz) BOLD signals from spatially distinct locations when no task was performed (*Biswal et al., 1995*), and over 4000 subsequent papers have documented different aspects of resting state fMRI. Importantly, these correlations have been widely adopted to infer functional connectivity between cortical regions (*Greicius et al., 2003*; *Fox et al., 2005*; *Smith, 2012*). The identification of patterns of highly correlated low-frequency signals in the resting brain provides a powerful approach to delineate and describe neural circuits, and an unprecedented insight into the manner in which distributed regions work together to achieve specific functions (*Pizoli et al., 2011*). In this study, we present the first robust demonstrations that similar phenomena can be detected within the gray matter of the human spinal cord, and we report our preliminary attempts to perform resting state connectivity studies within the cords of normal volunteers.

Although the vast majority of fMRI studies have explored function in the cerebrum, there have been a few investigations of function in the human brainstem and spinal cord. fMRI in the spinal cord was first performed (*Yoshizawa et al., 1996*) in 1996, and task-based (motor and/or sensory) spinal fMRI has since been demonstrated by a handful of groups worldwide (*Stroman et al., 1999*; *Backes et al., 2001*; *Madi et al., 2001*; *Giove et al., 2004*; *Moffitt et al., 2005*; *Maieron et al., 2007*; *Giulietti et al., 2008*;

*For correspondence:
robert.l.barry@vanderbilt.edu

**Competing interests:** The authors declare that no competing interests exist.

**Reviewing editor**: Timothy Behrens, Oxford University, United Kingdom

**eLife digest** Brain imaging methods such as functional magnetic resonance imaging (fMRI) can provide us with a picture of what the brain is doing when a person is carrying out a specific task. For example, an fMRI scan recorded whilst someone is reading is likely to show activity in regions in the left hemisphere of the brain that are known to be involved in language comprehension. fMRI can also be used to measure patterns of neuronal activity when someone is awake but not engaged in a specific task. This approach, known as resting state fMRI, can be used to examine which regions of the resting brain are active at the same time. Researchers are interested in these patterns of brain activity because they reflect neural circuits that work together to produce different functions and behaviors.

Over 4000 papers have used resting state fMRI to study the human brain. However, to date there has been no conclusive investigation of resting state activity in the spinal cord. This is largely because the spinal cord is much smaller than the brain, and most fMRI scanners are not sensitive enough to study it in detail. Consequently, little is known about intrinsic neural circuits in the resting spinal cord.

Now Barry et al. have used advances in fMRI technology to show that resting state functional connectivity does indeed exist in the spinal cord. Correlations were found in the resting levels of activity between spatially distinct areas of the cord, specifically between the ventral horns and between the dorsal horns. The ventral horns relay motor signals to the body, whilst the dorsal horns receive sensory signals from the body.

These findings also have clinical applications. Some patients with incomplete spinal cord injuries can recover near normal function, but the mechanisms responsible for this recovery are unclear because clinicians have not been able to probe neuronal connections in the spinal cord in a non-invasive manner. The work of Barry et al. should help with efforts to understand the neuronal changes that support recovery from spinal cord injury.

---

*Agosta et al., 2009a*; *Cohen-Adad et al., 2010*; *Summers et al., 2010*; *Brooks et al., 2012*; *Sprenger et al., 2012*). Spinal cord fMRI has primarily been used to study motor and sensory/pain pathways in the healthy spinal cord, but has also been shown to be sensitive to changes in patients with spinal cord injury (*Stroman et al., 2002*, *2004*; *Kornelsen and Stroman, 2007*) and multiple sclerosis (*Agosta et al., 2008a*, *2008b*, *2009b*; *Valsasina et al., 2010*, *2012*). Importantly, these spinal fMRI studies were focused on understanding spinal cord function when performing tasks, and to date only one paper has reported an investigation of resting state BOLD fluctuations in the human spinal cord, from which the results were equivocal (*Wei et al., 2010*). Partly, the lack of positive reports may reflect the relatively poor signal-to-noise ratio of spinal cord images achievable at conventional field strengths (1.5 Tesla and 3.0 Tesla) and the inherent limitations of low spatial resolution in studying small structures. The advent of ultra-high magnetic fields (7 Tesla and above) and implementation of appropriate multichannel spinal cord coils, along with improved image acquisition and correction protocols, provides new opportunities for high-resolution fMRI of the spinal cord with increased sensitivity to BOLD fluctuations in the small gray matter structures that are typically not well visualized at lower fields.

The spinal cord is essentially a long, cylindrical neural structure responsible for relaying motor and sensory information between the brain and body, it sits within a bath of cerebrospinal fluid (CSF), and is surrounded by large vertebral bodies and intervertebral discs (*Figure 1A*). A butterfly-shaped gray matter structure surrounded by densely packed white matter is found within the cord (*Figure 1B*). The gray matter is anatomically described by ventral (anterior), lateral, and dorsal (posterior) horns, though the lateral and ventral horns in thoracic and lumbar segments are often summarized as anterior (ventrolateral) gray matter. The dorsal horn contains neurons that receive sensory information from the extremities while upper motor neurons synapse onto lower motor neurons in the ventral horn, which relay information to the extremities (*Kandel et al., 2000*).

The relatively small number of spinal fMRI studies to date and lack of well-developed investigative tools make it difficult to formulate clear hypotheses of what low-frequency (< 0.08 Hz) BOLD signal correlations may be expected between sub-regions of spinal cord gray matter in a resting state. However, the formulation of such hypotheses may, in part, be guided by known anatomical connections or

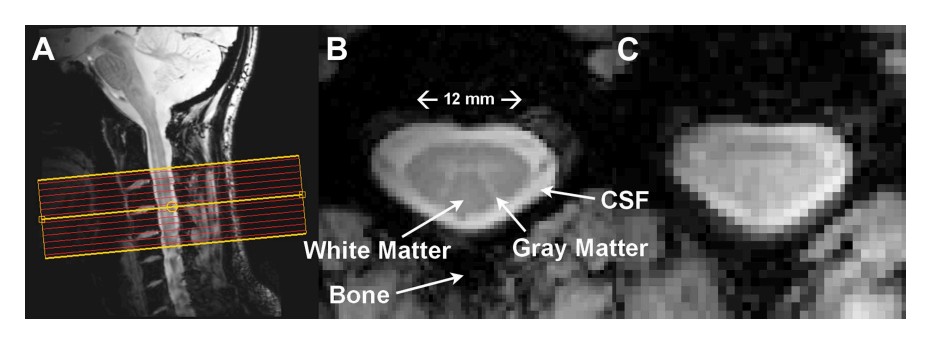

**Figure 1**. Resting state spinal cord fMRI at 7 Tesla. (**A**) Mid-sagittal slice from a healthy volunteer showing the complete cervical cord and typical axial slice placement for this resting state study. In all subjects the imaging stack was centered on the C3/C4 junction, providing full coverage of C3 and C4 and partial coverage of C2 and C5. (**B**) $T_2^*$-weighted anatomical image at C4 acquired with $0.6 \times 0.6 \times 4$ mm$^3$ voxels and interpolated to $0.31 \times 0.31 \times 4$ mm$^3$. Excellent contrast permits visualization of the characteristic butterfly-shaped gray matter column. (**C**) A single $T_2^*$-weighted functional image of this axial slice (acquired with $0.91 \times 0.91 \times 4$ mm$^3$ voxels). Functional images are high quality with minimal geometric distortions and $T_2^*$ blurring and permit adequate spatial delineation between white matter and cerebrospinal fluid.

spinal cord function. For example, the likely existence of central pattern generators in the human spinal cord that subserve basic locomotion (*Kandel et al., 2000*, p. 753) suggest that functional connectivity may exist between ventral (motor) horns. Similarly, reflexes suggest connections between a dorsal (sensory) horn and both ipsilateral and contralateral ventral horns (*Kandel et al., 2000*, Figure 36-2), and are primarily apparent in the presence of noxious stimuli. The ascending sensory pathways and descending motor pathways also suggest that there may be connectivity along the length of the cord, at least within individual dermatomes for dorsal horns (*Kandel et al., 2000*, p. 445). However, it must be emphasized that the lack of a direct anatomical connection between two sub-regions of spinal gray matter does not preclude the possibility of finding connectivity between these regions because they may be indirectly connected via other pathways. In practice, even if significant low-frequency signal variations related to function are manifest, they may be obscured by cord motion and various other sources of physiological noise. Here we adapt the paradigm used for earlier investigations of connectivity in the brain whereby we define very small regions of interest in anatomically distinct parts of spinal gray matter which in general subserve defined functions, and examine interregional steady-state correlations between them. In addition, we derive the patterns of voxels that show significant temporal correlation with selected single voxels within regions. These approaches have been successfully used, for example, in the cortex to delineate motor circuits. The use of high resolution images at 7 Tesla permit the reliable separation of ventral, dorsal, and bilateral segments of the cord so we can examine functional connectivity between sub-regions guided by known anatomical features. Functional connectivity along the cord may also be examined by considering any subregion in one slice and the same or another subregion in adjacent (or other) slices, but here we limit the group analysis of our first report to functional connectivity assessed within axial slices.

## Results

Functional images were preprocessed to mitigate rigid-body motion and physiological noise, and spatially interpolated to match the digital resolution of the $T_2^*$-weighted anatomical images (*Figure 2*). A 14-step standardized analysis protocol (described in 'Materials and methods') was used for each of the 22 subjects studied. In each subject, temporal signal-to-noise ratio (TSNR) was measured in spinal gray matter upon completion of the functional-to-anatomical affine registration (step #9) as well as after the application of CSF and white matter 'regressors of no interest' (steps #11 and #12). Across all 22 subjects, we observed a 30% increase in median TSNR (from 29.3 to 38.1) after the application of these few regressors, demonstrating the importance of characterizing and removing structured noise sources (*Xie et al., 2012*). After band-pass filtering to isolate the frequency range of interest

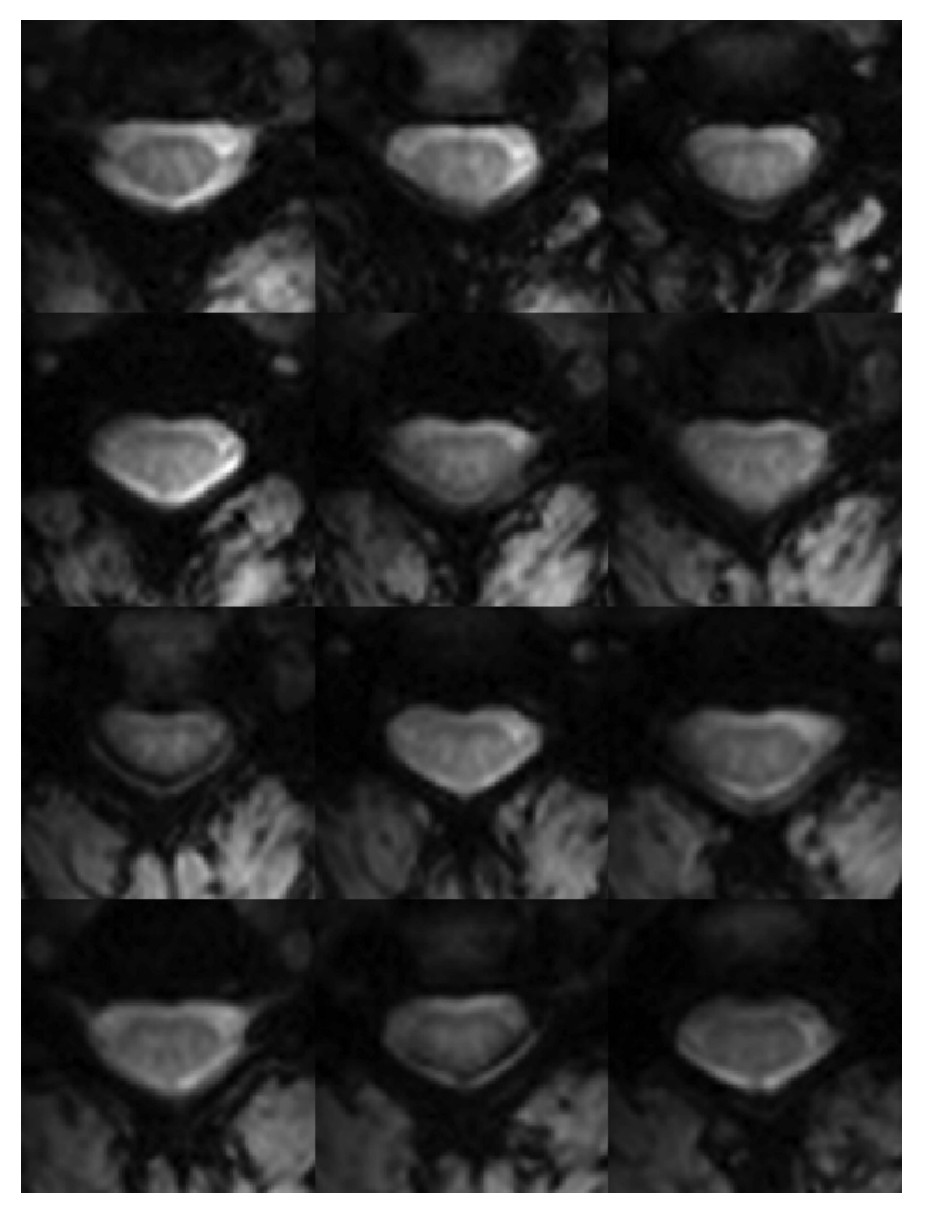

**Figure 2**. Functional weighted spinal cord images at 7 Tesla. A single volume of twelve contiguous $T_2$*-weighted slices centered on the C3/C4 junction (as illustrated in *Figure 1A*) in one subject. Each volume was acquired with $0.91 \times 0.91 \times 4$ mm$^3$ voxels and resampled to $0.31 \times 0.31 \times 4$ mm$^3$ voxels during the affine functional-to-anatomical registration. Excellent contrast between white matter and cerebrospinal fluid facilitates accurate registration between such functional volumes and high-resolution anatomical images (*Figure 1B*). The use of a 3D acquisition sequence with relatively short echo time and relatively few k-space lines per radiofrequency pulse provides high-quality images with minimal signal drop-out and geometric distortions, although artifacts caused by fat shift of the nerve root sleeve in the phase-encode direction still affect the dorsal edge in a few slices.

(0.01–0.08 Hz), single-subject analyses show that statistically significant correlations are measurable between selected regions and are reproducible across subjects. As an illustrative example with the corresponding time series, an analysis performed on one subject (female, 23 years old) demonstrates connectivity with the contralateral ventral horns in the same slice and with adjacent slices when a seed region is selected in the center of the right ventral horn (*Figure 3*). A stringent threshold of $|z| > 3.29$ (a two-tailed 99.9% confidence interval) was selected to show that connectivity is focused in the gray

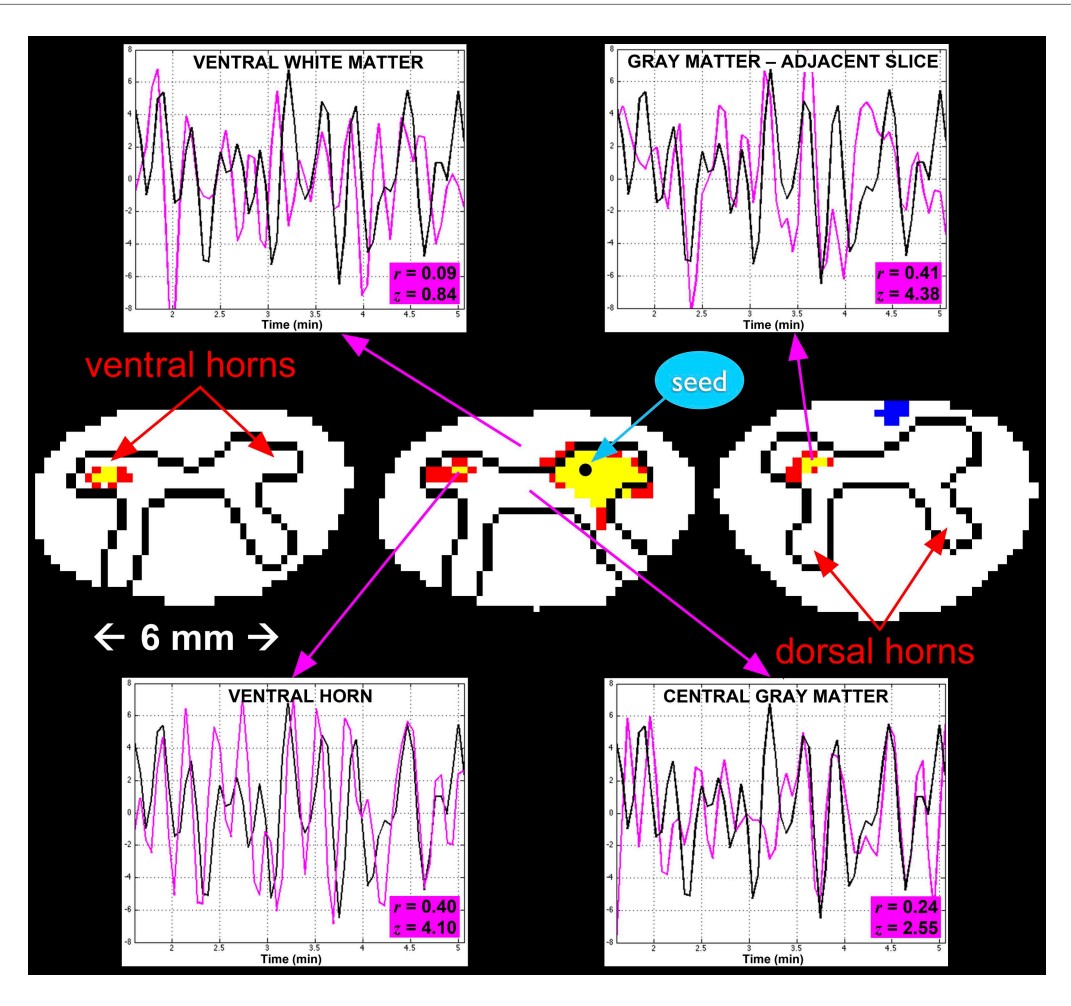

**Figure 3**. A single-subject analysis of resting state functional connectivity with corresponding time series. For clarity, only outlines of the gray matter butterfly and surrounding white matter are shown (rostro-caudal from left to right). Red and yellow represent statistically significant positive correlation with the seed time series (using a two-sided 99.9% confidence interval where red is 3.29 < z ≤ 3.89 and yellow is z > 3.89), and blue represents negative correlation (z < −3.29). The seed voxel is selected in the right ventral horn in C5, and exhibits functional connectivity with the contralateral ventral horn in the same slice as well as adjacent slices. Such connectivity between ventral horns is observed across all subjects. In each of the four plots, a 3.5-min segment of the seed time course is shown in black and the time course of the corresponding region of interest is shown in magenta. The highest correlations are observed in the contralateral ventral horn on the same slice (z = 4.10) and on the adjacent slices (z = 4.38). Correlations with central gray matter (z = 2.55) and adjacent white matter (z = 0.84) are relatively low, which, given the small size of the spinal cord, suggest that such correlations are genuine and not dominated by widespread physiological noise.

matter horns and not in central gray matter (connecting left and right sides and largely dominated by the central canal) nor adjacent white matter, which provides evidence that such gray matter correlations cannot be simply attributed to spatially correlated physiological noise and more likely represent genuine functional connectivity. Further examples of within-slice connectivity analyses in single subjects confirm that reproducible focal connectivity is found between ventral horns (*Figure 4A–F*) and between dorsal horns (*Figure 4G–J*). There is also evidence of plausible connectivity with central gray matter (*Figure 4K*) and between ventral and dorsal horns (*Figure 4L*), but these correlations are less consistent across all slices and not statistically significant at the group level. To quantify the occurrence of within-slice correlations between gray matter sub-regions across slices, we averaged time courses within each respective gray matter sub-region

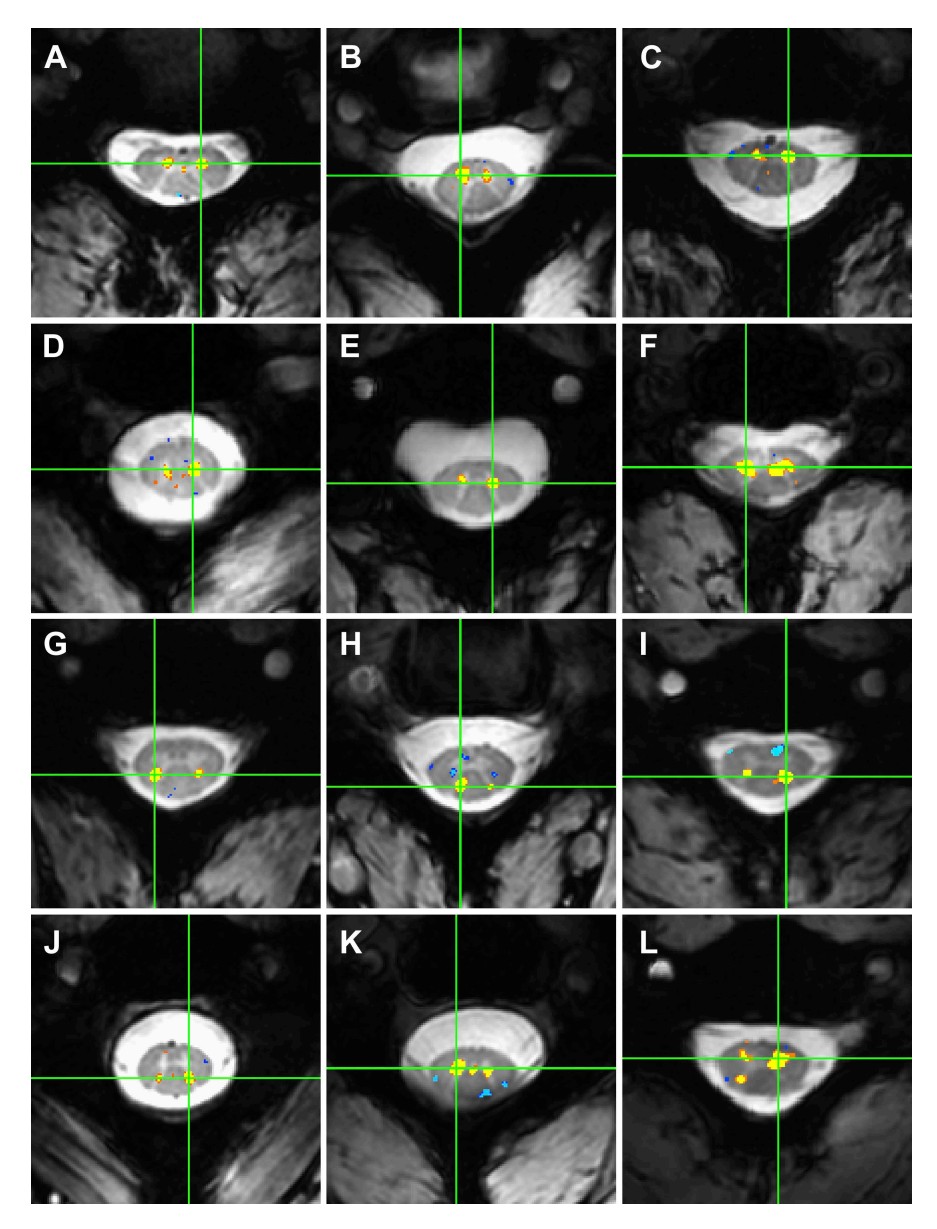

**Figure 4**. Examples of within-slice resting state functional connectivity across subjects. These analyses were performed using AFNI's 'InstaCorr' with p < 0.001 and no minimum cluster size. In each panel, a seed voxel is marked with a green crosshair and resultant correlations are overlaid on the anatomical image. (**A**)–(**F**) Connectivity between ventral horns for subjects 1, 3, 8, 10, 11, and 13, respectively. (**G**)–(**J**) Connectivity between dorsal horns for subjects 5, 16, 18, and 22, respectively. (**K** and **L**) Less common correlations within gray matter. In (**K**) (subject 20), focal connectivity between ventral horns and with central gray matter. In (**L**) (subject 7), connectivity between ventral horns but also with the contralateral dorsal horn. At the single-subject level, there is some evidence for functional connectivity between ventral and dorsal horns, but such correlations are less common across slices and not statistically significant at the group level.

(defined in step #14) and considered only positive correlations at a more conventional 95% confidence interval (z > 1.65; one-tailed). Across all 264 slices (12 slices/subject × 22 subjects), we observed that 67% of slices (177 of 264) exhibit significant correlations between ventral horns and 37% of slices (97 of 264) exhibit significant correlations between dorsal horns. In comparison, a markedly fewer number of slices (only 1 in 5) exhibited significant correlations between the remaining four pairs: 21% between left ventral and left dorsal horns (55 of 264), 21% between left

ventral and right dorsal horns (55 of 264), 20% between right ventral and left dorsal horns (54 of 264), and 23% between right ventral and right dorsal horns (62 of 264).

A group-level analysis of functional connectivity between sub-regions of spinal cord gray matter and adjacent white matter confirmed that the most robust correlations are observed between left and right ventral (motor) horns (p < 0.01; corrected), as well as between left and right dorsal (sensory) horns (p < 0.01; corrected) (*Figure 5A*). No significant group-level correlations (p > 0.05) were observed between other gray matter sub-regions, nor between spinal cord gray and white matter. Weak positive correlations were observed between left and right dorsal column white matter (p < 0.05; corrected), and negative correlations were observed between left ventral white matter and right

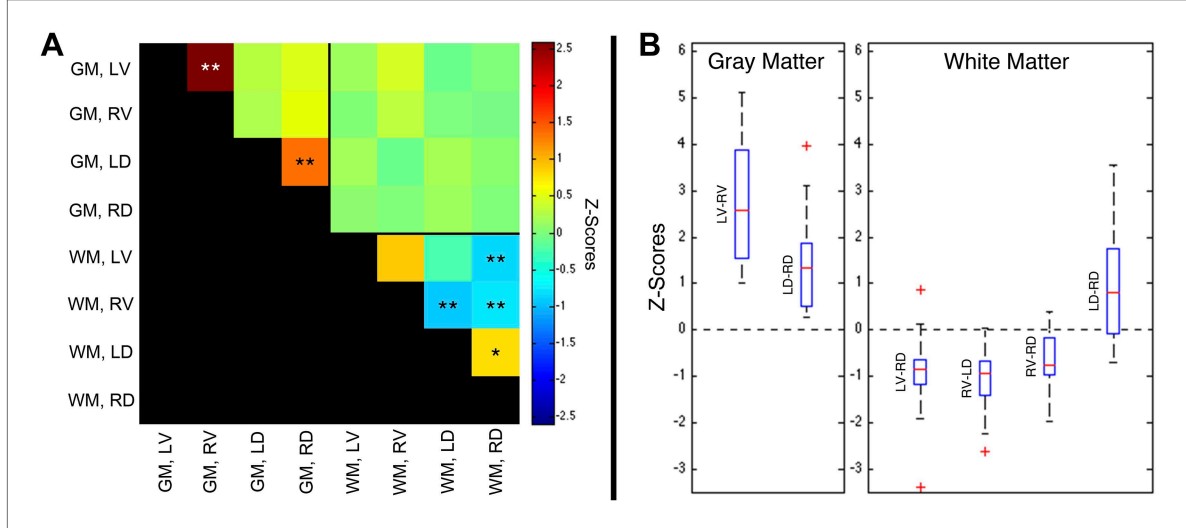

**Figure 5**. Group-level functional connectivity between sub-regions of spinal gray matter (GM) and surrounding white matter (WM) within slices. (**A**) In GM, strong positive correlations are observed between left (LV) and right (RV) ventral horns, as well as left (LD) and right (RD) dorsal horns (*p<0.05; **p<0.01; Bonferroni corrected). Weaker positive and negative correlations are observed within WM. No statistically significant correlations are observed between spinal GM and WM (upper right quadrant). (**B**) Box-and-whisker plots showing the median and upper and lower quartiles of the six statistically significant results identified in (**A**). Whiskers extend out to 1.5 times the distance between the upper and lower quartiles, and an outlier (beyond the whiskers) is denoted by a plus. Wilcoxon signed rank tests identify the distributions of z-scores (across all slices and subjects) that are significantly different from zero (p < 0.05). Functional connectivity between WM sub-regions is more variable and exhibits both positive and negative median correlations. In comparison, median GM correlations between LV-RV and LD-RD are positive across all 22 subjects. Additional analysis permutations (described in *Figure 5—figure supplements 1, 3, and 5* with example GM power spectra shown in *Figure 5—figure supplements 2, 4, and 6*) reveal that the three negative WM correlations are influenced by WM regression (step #12) and thus are open to more than one interpretation. However, supplementary analyses reveal that the positive GM correlations between ventral horns and between dorsal horns persist across all preprocessing permutations. These additional analyses further support the conclusion that positive correlations between GM horns are not artifactual—possibly created by preprocessing choices or frequency bandwidth selection—and most likely represent genuine functional connectivity.

The following source data and figure supplements are available for figure 5:

**Source data 1**. Matlab file containing the raw data used to generate *Figure 5*.

**Figure supplement 1**. Functional connectivity matrices resulting from preprocessing pipeline permutations.

**Figure supplement 2**. Power spectra across gray and white matter sub-regions for data filtered between 0.01 and 0.08 Hz.

**Figure supplement 3**. Functional connectivity matrices resulting from preprocessing pipeline permutations after band-pass filtering between 0.01 and 0.07 Hz.

**Figure supplement 4**. Power spectra across gray and white matter sub-regions for data filtered between 0.01 and 0.07 Hz.

**Figure supplement 5**. Functional connectivity matrices resulting from preprocessing pipeline permutations after band-pass filtering between 0.01 and 0.13 Hz.

**Figure supplement 6**. Power spectra across gray and white matter sub-regions for data filtered between 0.01 and 0.13 Hz.

dorsal white matter (p < 0.01; corrected), and between right ventral white matter and both left (p < 0.01; corrected) and right (p < 0.01; corrected) dorsal white matter. The apparent existence of negative correlations in resting state spinal cord data is not unexpected because anticorrelations are commonly observed in resting state analyses of the brain (*Chang and Glover, 2010*) and have been a topic of intense discussion for over a decade. The ranges of values within these six statistically significant distributions are presented as box-and-whisker plots (*Figure 5B*). The lower quartile is above zero in both gray matter plots, demonstrating that positive gray matter connectivity is a robust and reproducible measurement. In comparison, temporal correlations between white matter sub-regions are more variable and exhibit both positive and negative median correlations. The raw data used to generate this figure is provided as *Figure 5—source data 1*. Additional analyses were performed (*Figure 5—figure supplement 1*, *Figure 5—figure supplement 3*, and *Figure 5—figure supplement 5*) to confirm that positive gray matter correlations are stable across various preprocessing procedures whereas white matter correlations are positive before but negative or non-significant after white matter regression (step #12). These supplementary analyses also showed that weaker positive correlations between sub-regions of left and right dorsal white matter remained positive and significant across various preprocessing configurations.

## Discussion

We have presented the first conclusive demonstration that ultra-high field fMRI can non-invasively detect and characterize resting state BOLD signals within the gray matter of the human spinal cord. Low-frequency temporal correlations were observed between gray matter horns in all subjects, and examples of these correlations are presented in *Figure 3* and *Figure 4*. Within a given axial slice, the strongest and most robust correlations were between left and right ventral horns. Correlations were also observed between left and right dorsal horns, and the reproducibility of correlations between ventral horns and between dorsal horns was demonstrated within a cohort of 22 healthy volunteers (*Figure 5*). Although correlations between ventral and dorsal gray matter were also observed at the single-subject level (*Figure 4L*), such findings were less frequent and not statistically significant at the group level. The absence of group-level correlations between spinal gray matter and adjacent white matter showed that these positive gray matter correlations are unlikely to be driven by spatially-correlated physiological noise. In fact, supplementary analyses without CSF or white matter regressors (*Figure 5—figure supplement 1a*) revealed strong correlations within gray matter and within white matter but not between gray and white matter, suggesting that gray and white matter in the spinal cord may exhibit different degrees of physiological fluctuations. The main analyses (*Figure 5*) also showed anticorrelations between white matter sub-regions. These observations were not unexpected because negative correlations are commonly seen in the brain (*Chang and Glover, 2010*), but additional analyses (*Figure 5—figure supplement 1*) revealed that correlations within white matter were heavily influenced by preprocessing methodology. As a result, the nature of negative white matter correlations remains unclear and requires further investigation. However, predominantly positive correlations between dorsal white matter sub-regions persisted across preprocessing permutations. Although the origins of these positive white matter correlations remain to be determined, and whereas in general BOLD signals from activation have been difficult to detect in brain white matter, we note that Ding et al. recently reported the reliable detection of anisotropic correlations of resting state BOLD signals in brain white matter that appear to mimic white matter tracts identified by diffusion imaging methods (*Ding et al., 2013*). Moreover, the white matter sub-regions within the cord also tend to be close to draining veins.

Another observation is that z-scores measured between ventral horns tended to be higher than z-scores measured between dorsal horns—a finding that was highly significant (p < 0.01 using a two-tailed Wilcoxon signed rank test; 'signrank' in Matlab). There are several possible reasons for this finding. Firstly, the dorsal horns tend to be slightly narrower than the ventral horns, and thus may be more susceptible to registration inaccuracies, residual physiological noise, and partial volume averaging with adjacent white matter. Secondly, as shown in *Figure 2*, signal dropout and unavoidable artifacts affect the dorsal horns but not ventral horns in a few slices, which would bias the results in favor of ventral horn connectivity. Finally, even if spatial artifacts, registration inaccuracies, signal dropout, physiological noise, and partial volume averaging effects were minimized, it may be that functional connectivity between dorsal horns is more variable if the automated selection of gray matter sub-regions (step #14) isolate signals from different laminae within dorsal horns (*Ruscheweyh and Sandkühler, 2002*).

To date, clinical applications of task-based spinal fMRI studies have primarily been targeted to subjects with multiple sclerosis (*Agosta et al., 2008a, 2008b, 2009b*; *Valsasina et al., 2010, 2012*) and spinal cord injury (SCI) (*Stroman et al., 2002, 2004*; *Kornelsen and Stroman, 2007*). We propose that the non-invasive methods of resting state spinal cord functional connectivity developed in this paper may be most readily translatable to clinical investigations characterizing damage due to acute or chronic SCI and monitoring the efficacy of surgical or pharmacological interventions. Functional connectivity and the assessment of plasticity in the human spinal cord and animal models of SCI have been topics of intense research for many years (*Bregman et al., 1997*; *Raineteau and Schwab, 2001*; *Cai et al., 2006*; *Freund et al., 2011*) because SCI affects 260,000 people in the United States (a prevalence of ~1 in 1200) with 11,000 new injuries reported each year (*National Spinal Cord Injury Statistics Center, 2010*). Studies have investigated the role of propriospinal neurons in partial recovery from incomplete SCI (*Bareyre, 2008*; *Flynn et al., 2011a*) while in vitro analyses of functional connectivity in mouse spinal cord have relied on electrophysiological methods (*Flynn et al., 2011b*). A new intervention strategy using epidural stimulation and stand training was recently developed that has to date restored voluntary movement in four patients with complete paralysis, demonstrating that functional connectivity across a lesion may be restored with epidural stimulation (*Angeli et al., 2014*). Such studies would likely benefit from an ability to assess the functional architecture of the spinal cord throughout therapy. The majority of spinal injuries are, however, incomplete, and lost function may eventually return to near-normal levels; however, the progression of functional recovery after incomplete SCI remains poorly understood due to the absence of a non-invasive method to reliably assess spinal cord connectivity in vivo. We propose that resting state acquisitions of the cervical spinal cord will become a valuable tool for characterizing changes in functional connectivity in SCI, and for the prognosis and monitoring of progression of recovery via spontaneous repair and/or surgical intervention. For example, the imaging volume (shown in *Figure 1A*) may be centered on a focal injury to the cord, and functional connectivity above and below the injury may be assessed. This process with identical slice placement could then be repeated serially over time to investigate phenomena of neural plasticity and adaptation of spinal pathways. Moreover, even in normal subjects, the functional organization of the spinal cord is relatively under-explored and remains poorly understood. Observations of altered resting state connectivity in the brain in numerous disorders (*Fox and Greicius, 2010*) and as a function of behavior or cognitive skills suggest that such correlations reflect an important level of organization and may play a fundamental role in the execution and maintenance of various functions (*Pizoli et al., 2011*). Thus, investigation of resting state spinal cord networks could similarly have widespread applicability in studying central nervous system diseases that affect motor and/or sensory pathways such as cervical spondylotic myelopathy, neuromyelitis optica, acute disseminated encephalomyelitis, arachnoiditis, transverse myelitis, amyotrophic lateral sclerosis, and multiple sclerosis.

Our study has three main limitations. Firstly, although we observed statistically significant correlations along the cord in single-subject analyses (for example, in left ventral gray matter in *Figure 3*), we chose to constrain the group analysis to investigating connectivity within each axial slice. This was done because incorporating correlations along the cord would have made the analysis significantly more complicated by increasing the number of potential correlations by an order of magnitude. Furthermore, as shown in *Figure 2*, some slices exhibited regions of signal dropout at the dorsal edge caused by fat shift of the nerve root sleeve in the phase-encode (anterior-posterior) direction. The absence of a statistically significant correlation with a slice impacted by main field ($B_0$) inhomogeneities or an artifact cannot in and of itself be interpreted as proof that connectivity does not exist between particular sub-regions of interest, and ultimately very careful single-subject analyses will need to be performed to reliably characterize functional connectivity along the cord. Secondly, although we have investigated one aspect of reproducibility (via a group analysis of 22 healthy volunteers) and established the existence of spinal cord functional connectivity, further investigations of within-subject reproducibility still need to be performed. This could be done by acquiring multiple resting state runs from a single scanning session and/or by scanning the same volunteer on multiple days while ensuring that the imaging volume is consistently placed in the same location (e.g., centered on the C3/C4 junction, as shown in *Figure 1A*). The reproducibility of these connectivity measures over months and years will need to be quantified before such techniques can be reliably used to study disease progression or gerontology. Thirdly, this study reports empirical findings of functional connectivity in the spinal cord but does not directly address the physiological origins of these low-frequency BOLD signal fluctuations. However, as a point of reference, observations of functional connectivity were first reported

in the brain nearly two decades ago (*Biswal et al., 1995*) and yet the origin of these fluctuations is still a topic of intense discussion. Although the nature of resting state correlations is not fully understood, recent studies have combined tractography and functional imaging with techniques such as network analysis and graph theory to explore how these networks may have emerged (*Deco et al., 2013a*, *2013b*; *van den Heuvel and Sporns, 2013*; *Mišić et al., 2014*; *Goñi et al., 2014*). The origin of resting state networks has also been explored within the context of evolution and the expansion of the cerebral cortex (*Buckner and Krienen, 2013*). We therefore propose that because the spinal cord is an integral part of the central nervous system, one simple explanation is that low-frequency BOLD fluctuations in the brain and spinal cord share the same origin. This theory would then suggest that there may be long-range connections between networks in the spinal cord and cerebrum (and cerebellum), and future research should consider the nature of resting state networks not only within the brain but within the entire central nervous system.

Previous attempts to detect functional connectivity in the spinal cord have undoubtedly been confronted by significant technical challenges. At conventional field strengths (1.5 Tesla and 3.0 Tesla), the lower signal-to-noise ratio and BOLD contrast limit the spatial resolution and sensitivity for detecting BOLD signals, so large voxels are typically used and multiple acquisitions are averaged. At 7 Tesla, high resolution BOLD acquisitions of the spinal cord have not evolved as rapidly as for the brain in part due to the widespread dependence on single-shot echo-planar acquisitions at lower fields and the lack of specialized coils to image the spinal cord. We addressed these challenges and limitations by using a 7 Tesla scanner, novel fMRI image acquisition and data correction protocols, and a dedicated 16-channel radiofrequency coil array designed for cord imaging. Our results are the first demonstrations of functional imaging of the human spinal cord at 7 Tesla (*Barry et al., 2013a*) and high-resolution resting state functional connectivity in the spinal cord (*Barry et al., 2013b*), and are likely to be of significant relevance in understanding basic aspects of spinal cord function both in normal development and in clinical disorders of the central nervous system.

## Materials and methods

### Data acquisition

Experiments were performed on a Philips Achieva 7 Tesla scanner with a custom-designed (Nova Medical Inc.) quadrature transmit and 16-channel receive coil array for cervical spinal cord imaging. 22 healthy volunteers (11 male, 21–63 years; 11 female, 23–34 years; 28.4 ± 8.8 years) with no history of spinal cord injury or neurological impairment were recruited and scanned under protocols approved by the Institutional Review Board at Vanderbilt University Medical Center. Female participants of childbearing potential required a negative urine pregnancy test for the scan to proceed. Non-MR study data were collected and managed using REDCap electronic data capture tools hosted at Vanderbilt University (*Harris et al., 2009*). REDCap (Research Electronic Data Capture) is a secure, web-based application designed to support data capture for research studies.

Anatomical axial images with high spatial resolution and $T_2^*$-weighting (*Figure 1B*) were acquired with the following MR parameters: field of view = 160 × 160 mm$^2$, 12 4-mm slices (centered on the C3/C4 junction, as shown in *Figure 1A*), nominal voxel size = 0.6 × 0.6 × 4 mm$^3$, interpolated voxel size = 0.31 × 0.31 × 4 mm$^3$, repetition time = 303 ms, echo time = 8.2 ms, flip angle = 25°, sensitivity encoding (SENSE) (*Pruessmann et al., 1999*) reduction factor = 2.0 (anterior-posterior), signal acquisitions = 8, total acquisition time = 5 min 22 s.

Functional images with identical slice placement were acquired with a 3D multi-shot gradient-echo sequence (*van der Meulen et al., 1988*) previously shown to minimize $T_2^*$ blurring and geometric distortions in cortical fMRI at 7 Tesla (*Barry et al., 2011*). The functional MR parameters for the first 11 subjects were: field of view = 160 × 160 mm, twelve 4-mm slices, voxel size = 0.91 × 0.91 × 4 mm$^3$, repetition time = 18 ms, echo time = 7.8 ms, flip angle = 15°, echo train length = 9, SENSE reduction factor = 1.56 (anterior-posterior), volume acquisition time = 3.6 s (300 ms/slice), number of volumes = 150 (after 10 'dummy' scans), total scan time = 9 min 38 s, max gradient strength = 30 mT/m, max slew rate = 175 mT/m/ms. The functional MR parameters for the last 11 subjects were the same except for the following minor adjustments: repetition time = 17 ms, echo time = 8.0 ms, volume acquisition time = 3.34 s (278 ms/slice), total scan time = 9 min. For all subjects, respiratory and cardiac cycles were externally monitored and recorded using a respiratory bellow (placed on the abdomen) and pulse oximeter (placed on the left index finger).

## Data processing

Functional data were corrected for physiological noise using methods that are commonly used in fMRI of the brain in addition to novel data-driven 'regressors of no interest'. The steps applied to all spinal fMRI data are as follows:

1.  For each slice of anatomical and functional images, a 2D Gaussian weighting kernel was manually defined with the full-width-at-half-maximum set at the CSF boundaries. Weighting masks defined on anatomical images were used in affine registration (step #8), and weighting masks defined on functional images were used in affine registration as well as rigid-body motion correction (step #5).
2.  For each slice, a 'not-spine' mask was defined by drawing a region around the entire spinal cord and then logically inverting it (used in step #3).
3.  For each slice, data-driven 'regressors of no interest' were selected via principal component analysis (PCA) of all voxels within the not-spine mask to identify structured noise sources that would similarly affect the spinal cord and external (neck) regions. The number of eigenvectors selected reflected up to 80% of the slice-wise cumulative variance or until the difference between two successive eigenvalues was less than 2% (typically 3–5 per slice). These vectors were regressed from the time series of all voxels within a slice, and significantly improved the efficacy of motion correction (step #5) by mitigating widespread intensity fluctuations due to physiological processes (e.g., swallowing).
4.  For each slice, a representative (target) volume was automatically selected for motion correction by calculating the median intensity of each voxel (over time) and then selecting the volume closest to the median image (identified via minimal least squares error).
5.  Rigid-body motion correction was performed on a slice-wise basis (using 3dWarpDrive in AFNI [*Cox, 1996*]) using the target volumes (identified in step #4). Motion was constrained to be within-plane translation (i.e., no rotation of the spinal cord). To mitigate the detrimental effects of sporadic artifacts (e.g., swallowing) on motion parameter estimation, translation estimates were filtered with a 5-point median filter and then re-applied (using 3dAllineate in AFNI [*Cox, 1996*]) to the original data before motion correction. The initial registration (to obtain motion parameter estimates) used quintic interpolation and the final transformation used sinc interpolation.
6.  An established image correction technique called RETROICOR [*Glover et al., 2000*] (implemented in AFNI [*Cox, 1996*]) was applied to the entire functional volume to further reduce quasi-periodic intensity variations due to physiological noise.
7.  Using the high-resolution anatomical images as a reference, masks defining the boundaries of gray matter, white matter, and CSF were created for each slice.
8.  Affine registration of the target volume (identified in step #4) to the anatomical image was performed (via 3dAllineate) on a slice-wise basis. The Hellinger metric was selected as the cost function, and the degrees of freedom were constrained to within-plane translation, scaling (maximum of 1% in the read direction and 5% in the phase-encode direction), and shearing (maximum of 5%).
9.  The affine transforms (defined in step #8) were applied to all functional volumes (via 3dAllineate), and transformed functional images were resampled (with sinc interpolation) to match the final resolution of the anatomical volume (voxel size = 0.31 × 0.31 × 4 mm$^3$).
10. The quality of the final functional-to-anatomical alignments was visually verified using MRIcron (www.mccauslandcenter.sc.edu/mricro/mricron).
11. For each slice, additional data-driven 'regressors of no interest' were selected via PCA of all functional voxels within the CSF mask (defined in step #7) to identify structured noise sources that would similarly affect gray matter and CSF. The number of eigenvectors selected reflected up to 50% of the slice-wise cumulative variance or until the difference between two successive eigenvalues was less than 2% (typically 2–6 per slice). These vectors were regressed from the time series of all spinal cord voxels within a slice.
12. For each slice, a 'global' white matter signal was calculated via PCA of all functional voxels within the white matter mask and extraction of the first eigenvector (typically representing 10–30% of the variance). This was primarily done to mitigate any residual variance due to shifting of the white matter boundary (caused by motion) but would also reduce variance caused by residual physiological noise. This vector was regressed from the time series of all gray and white matter voxels within a slice.
13. In preparation for analyses of functional connectivity, resultant functional data were band-pass filtered between 0.01 and 0.08 Hz using a Chebyshev Type II filter ('cheby2' and 'filtfilt' in Matlab) to emphasize low-frequency signals of interest.

14. In preparation for group analyses, gray and white matter masks (defined in step #7) were subdivided into quadrants to identify left and right ventral and dorsal horns (excluding central gray matter connecting left and right sides), as well as the four adjacent white matter regions. Each of these eight sub-region masks (per slice) was morphologically eroded (using 'imerode' in Matlab) to remove the outermost voxels and mitigate partial volume effects. For gray matter the morphological eroding object was a disk with a radius of 3 voxels, and for white matter it was a disk with a radius of 11 voxels. If the eroded sub-region did not contain any voxels (i.e., the disk was too large) then the disk size was incrementally decreased by 1 voxel, and the erosion process repeated until the innermost area of each sub-region was extracted. Time series extracted from each eroded sub-region were also used in single-subject analyses (except for in *Figure 3* and *Figure 4*, which used single-voxel correlations) and power spectra calculations (figure supplements for *Figure 5*).

Figure supplements for *Figure 5* investigate modifications to steps #11, #12, and/or #13, and deviations from this standardized preprocessing pipeline are described in the respective figure legends.

## Data analysis

For the voxel-based analysis shown in *Figure 3*, the linear correlation coefficient ($r$) was calculated between a seed voxel and all other voxels in Matlab. These correlation values were converted to $z$-scores using the Fisher $r$-to-$z$ transformation $z = \tanh^{-1}(r)(dof–3)^{1/2}$ where $dof$ is the estimated degrees of freedom for each voxel after correction for first-order autocorrelation (*Rogers and Gore, 2008*). A statistical threshold of $|z| > 3.29$ (a two-tailed 99.9% confidence interval) was selected to clearly show that gray matter correlations tend to be focused in the center of the horns. A minimum cluster threshold of nine contiguous interpolated voxels ($3.49$ mm$^3$, approximately equal to the native functional voxel volume of $3.31$ mm$^3$) was also used to further protect against spurious correlations.

For the single-subject analyses presented in *Figure 4*, AFNI's 'InstaCorr' (*Cox, 1996*) was used to display correlations between a single gray matter voxel and all other voxels within the spinal cord. The correlation threshold was $p < 0.001$ and in these analyses no cluster thresholding was used to reveal all correlations within gray and white matter.

Finally, for the group-level region-of-interest (ROI)-based analyses (*Figure 5*), the time series of individual voxels were averaged within each eroded mask (created in step #14) for each subject. The linear correlation coefficient was then calculated between the averaged time course from a sub-region and all other sub-regions. These correlation coefficients were converted to $z$-scores and corrected for first-order autocorrelation (*Rogers and Gore, 2008*). For each ROI comparison (e.g., left dorsal horn vs right dorsal horn), the median $z$-score (unthresholded) was calculated across all 12 slices for each of the 22 subjects, and then a two-tailed Wilcoxon signed rank test ('signrank' in Matlab) identified group-level distributions of 22 $z$-scores that were significantly different from zero ($p < 0.05$ or $p < 0.01$ with a Bonferroni correction factor of 28).

## Acknowledgements

The authors thank David S Smith for assistance with Matlab code optimization, and Benjamin N Conrad for assistance with subject recruitment. This research was primarily supported by NIH grants 1K99EB016689-01A1 and 1R21NS081437-01A1, but also by grants 5R01EB000461, S10RR023047, and 5K01EB009120. The project was also supported by the National Center for Research Resources, grant UL1RR024975-01, which is now at the National Center for Advancing Translational Sciences, grant 2UL1TR000445-06. The content is solely the responsibility of the authors and does not necessarily represent the official views of the NIH.

## Additional information

### Funding

| Funder | Grant reference number | Author |
| --- | --- | --- |
| National Institutes of Health | 1R21NS081437-01A1 | Robert L Barry, John C Gore |
| National Institutes of Health | 5R01EB000461 | John C Gore |

| Funder | Grant reference number | Author |
|---|---|---|
| National Institutes of Health | 5K01EB009120 | Seth A Smith |
| National Institutes of Health | 1K99EB016689-01A1 | Robert L Barry |
| National Institutes of Health | UL1RR024975-01 | Robert L Barry, Adrienne N Dula |
| National Institutes of Health | 2UL1TR000445-06 | Robert L Barry, Adrienne N Dula |
| National Institutes of Health | S10RR023047 | John C Gore |

The funders had no role in study design, data collection and interpretation, or the decision to submit the work for publication.

## Author contributions

RLB, SAS, Conception and design, Acquisition of data, Analysis and interpretation of data, Drafting or revising the article; AND, JCG, Conception and design, Analysis and interpretation of data, Drafting or revising the article

## Ethics

Human subjects: Healthy volunteers with no history of spinal cord injury or neurological impairment provided informed consent to be scanned, and to have the research published, under protocols 111087 and 140202 approved by the Institutional Review Board at Vanderbilt University Medical Center.

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
