## [Decision Letter]

Thank you for sending your work entitled “Resting state functional connectivity in the human spinal cord” for consideration at *eLife.* Your article has been favorably evaluated by Eve Marder (Senior editor) and 2 reviewers, one of whom, Timothy Behrens, is a member of our Board of Reviewing Editors.

The Reviewing editor and the other reviewer discussed their comments before we reached this decision, and the Reviewing editor has assembled the following comments to help you prepare a revised submission.

We were impressed with the new cohort of subjects, and thought the manuscript substantially improved, but we still had several major technical issues with the data; Figure 4 in particular, that prevent us from recommending publication in the current form. I have summarised the reviews below. Both reviewers had major queries about the statistical treatment and data presentation. We hope that you will be able to address these comments in revision. We note that we regard the statistical issues detailed in points 1-3 below as essential.

The issues surround the analyses in Figure 4.

1) The main result in (a) is a matrix of correlations, only some of which are shown (the others are zeroed if non-significant).

The reviewers had two major issues with this matrix:

(a) “This figure indicates that the z-score for e.g. the upper-right 4x4 square is exactly zero (light blue color) for each of the 16 correlations. The authors state in their response that they just wanted to show significant correlations (and they thus set all the non-significant ones to zero), but in my opinion this is obscuring part of the results. In the hypothetical worst case, all of these 16 correlations might be just below significance, which would call into question the specificity of their results. I'm not at all trying to suggest that this is the case but I would guess that the readers would like to judge for themselves how convincing the data are and it is thus essential to show these correlations as well, especially since spinal fMRI is still regarded with some scepticism. I am aware that the authors supplied the data that underlie Figure 4, but I couldn't analyse them, as none of my browsers allowed a correct download of the .mat file.”

But more importantly:

(b) “Almost all the correlations that are expected to be zero include the white matter. It is frankly ridiculous to regress out the white matter signal in pre-processing, and then say that there is no functional correlation within white matter. This is surely guaranteed to be the case. Without the white matter squares though, this figure would be simply two numbers, which wouldn't make much of a figure.”

The second set of important comments pertains to the group analysis in Figure 4.

“The statistical tests in b are not sensible. All data are averaged across slices and subjects, artificially increasing degrees of freedom, reducing standard errors and leading to completely unreasonable p-values. This is clearly unreasonable. The correct analysis averages across slices within subject, and then performs the statistical test across subjects like any normal ROI-based analysis. There are several reasons why this analysis is not reasonable, but the most obvious is that the slices are not independent. Others include the fact that even if the slices were independent and heteroschedastically distributed, the nature of the random effect that would be inferred is not the population of individuals, but rather the population of slices.”

And:

“Despite the authors' arguments in their response, I still argue that it is not valid at the group level to test across slices and subjects. They should average across slices so that they test a set of 19 data-points (corresponding to the number of subjects) against zero, and not 228 (corresponding to the number of subjects times the number of slices), as they still do.”

After discussion, we would like to see the traditional analysis. If possible we would like to see the results of the simplest possible analysis which is a t-test on the first-level z-scores, as well as the non-parametric analysis proposed (but with the new slice averages, rather than the original slices).

3) We think it very odd that Figure 4 does not contain any spatial or spectral evidence at the group level.

(a) If possible, we would therefore like to see the group average spatial maps of the correlations for seed points in the 2 dorsal and 2 ventral horns. Again, the average should be performed within subject first, and then significance should be determined across subjects.

(b) We would like to see the power spectra for the time-series of the “resting functional connectivity”. We believe this is important information, given what is known about these spectra for cortical resting activity.

Two further comments from review:

4) I am still confused as to how the seed voxel for each individual subject analysis in each gray matter horn was chosen. The authors nicely describe how this was done for the group analysis, but still give no information for the individual analyses.

5) The authors' new choice of threshold for the group analysis (previously: p < 0.01, now: p < 0.000001) seems to be a bit arbitrary. They could just use p < 0.05 with Bonferroni or FDR correction for the number of tests (28 in this case). In my eyes, this would be a more appropriate and well-motivated threshold.

[Editors’ note: a previous version of this study was rejected after peer review, but the authors submitted for reconsideration. The previous decision letter after peer review is shown below.]

Thank you for choosing to send your work entitled “Resting state functional connectivity in the human spinal cord” for consideration at *eLife*. Your full submission has been evaluated by a Senior editor, 2 peer reviewers, and a member of our Board of Reviewing Editors (Timothy Behrens), and the decision was reached after discussions between the reviewers. We regret to inform you that we cannot offer to publish your work in *eLife* at this point.

In sum, we all agreed that the work was potentially of high interest, and that we would be in principle interested in publishing the work. However, we were compelled to reject the manuscript in its present form due to the unusually small size of the subject cohort, and doubts about the statistical robustness of some of the key results. We would like to emphasize that if you chose to significantly increase the cohort size, and performed some extra analyses, we would be happy to reconsider a new submission of the manuscript. In this case, it would not be subject to triage but would be reassigned to Tim Behrens, and it is highly likely he would consult the same reviewers (assuming they are willing).

Whilst we do not wish to prescribe precise requirement for cohort sizes at *eLife*, we draw your attention to work from, for example, Bertrand Thirion, which suggests that fMRI results in cortex begin to asymptote in terms of reproducibility at cohort sizes of approximately 20, and that those with cohort sizes of less than 10 were very unlikely to lead to reproducible results. It was the opinion of the reviewers that there is no reason why this would be different in spinal fMRI. The reviewers also had several suggestions to make the statistics more robust that you will see below. We do not wish to prescribe which of these approaches you choose to use, but if you chose to resubmit, we would certainly examine the statistical robustness of the new results with scrutiny.

The amalgamated comments of the reviews and the discussion between the reviewers can be found below:

*Praise for the paper*:

This is a highly interesting study. It is not only the first proper resting-state fMRI study of the spinal cord, but it is also the first spinal fMRI study in general at 7T. The authors seem to have overcome the problems of spinal imaging at ultra-high fields and have obtained functional images of remarkable quality. They have also developed a very thorough processing pipeline for spinal fMRI at 7T, which will be of great interest to groups working on brainstem and spinal imaging. The manuscript is well written and their demonstration of resting-state networks in the spinal cord is of potentially high clinical interest for various motor and sensory disorders. However, the study also has several weaknesses that need to be addressed.

This is a very well written and significant paper from a prominent imaging group. The hypothesis has been clearly presented and the results and discussion very well reported and discussed.

*Comments on the data*:

1) Why did they only scan such a low number of subjects (N=7; adding to the fact that there are only 150 scans per subject)?

2) The authors do not mention to what extent their various steps of data clean-up (removal of non-spine signals, motion correction, etc) decrease the noise in their time-series and thus allow reliable identification of resting-state networks. They should show the average spinal cord temporal signal to noise ratio before and after (each of) those steps.

3) It is unclear how the seed voxels for the individual analyses were chosen in each of the four spinal quadrants.

4) Their choice of threshold (p < .001 uncorrected with a one-voxel extent threshold) seems to be extremely lenient and why do they use a different threshold (p < .01) for the group analysis?

5) I am not sure about their combination of slices and subjects into one test - normally one would want to have one parameter estimate per subject in a group analysis that tests for one effect.

The results of the individual connectivity analyses need to be expanded.

1) They only show results for one subject and only for the right ventral horn. The very least the authors should do is to show the same subject with dorsal horn connectivity.

2) It would also be important to know (maybe in a supplemental table) how many subjects showed significant connectivity between the ventral (dorsal) horns, i.e. how reproducible this result is on a single subject level.

3) They do not quantify at all the levels of intra- vs. inter-segmental connectivity, which are both part of their hypotheses.

The group-level results appear very weak for the dorsal cord (i.e. Z-score of less than 1). I would suggest including only slices in the analysis that exceed a certain TSNR threshold.

Reliability in general: Because the spinal cord is susceptible to a number of factors including motion, CSF pulsation, it would be great if the authors could show some clear indication of the reliability of their result. It is possible that the data from the subject could be split into half and the correlation (that was estimated) could be reported for the 2 halves for similarity.

*Comments on the manuscript*:

The introductory paragraph that contains the authors' hypotheses is very weak, due to several reasons. 1) The citation of nothing more than a 1000-page general neuroscience textbook for supporting their hypothesis of dorsal horn spontaneous activity is not appropriate. They should spell out the “anatomical connections” and “existence of central pattern generators” and support them with appropriate citations. To my knowledge, central pattern generators do not exist in the adult dorsal spinal cord, so I would like to know on which basis the authors expected dorsal resting-state networks. 2) Based on what evidence do they expect a) that correlations exist within “a vertebral level” (when the spinal cord is clearly not organized vertebrally but segmentally) and b) that correlations “within a given gray matter horn should be detectable along the cord”?

The discussion completely lacks an explanation of the possible physiological origin of spinal resting-state networks. Important concepts such as a) the level of spontaneous activity in the spinal cord, b) uni- vs. bilateral excitatory/inhibitory responses under tasks, and c) anatomical pathways providing a structural substrate for the observed functional connectivity are all lacking.

---

## [Author Response]

*We were impressed with the new cohort of subjects, and thought the manuscript substantially improved, but we still had several major technical issues with the data;*
Figure 4
*in particular, that prevent us from recommending publication in the current form. I have summarised the reviews below. Both reviewers had major queries about the statistical treatment and data presentation. We hope that you will be able to address these comments in revision. We note that we regard the statistical issues detailed in points 1-3 below as essential*.

We understand and agree with the concerns surrounding the statistical analyses in the previous Figure 4 (now Figure 5), and believe that the current analyses properly address the issues of non-independence, correction for multiple comparisons, and reasonable thresholds for statistical significance. Since our previous resubmission we have also acquired data from 3 more subjects, so the current analyses are on a cohort of 22 healthy volunteers. We have also further refined the preprocessing pipeline by improving the select of a target volume for motion correction (step #4) and the motion correction itself (step #5).

*The issues surround the analyses in*
Figure 4.

*1) The main result in (a) is a matrix of correlations, only some of which are shown (the others are zeroed if non-significant)*.

*The reviewers had two major issues with this matrix*:

*(a) “This figure indicates that the z-score for e.g. the upper-right 4x4 square is exactly zero (light blue color) for each of the 16 correlations. The authors state in their response that they just wanted to show significant correlations (and they thus set all the non-significant ones to zero), but in my opinion this is obscuring part of the results. In the hypothetical worst case, all of these 16 correlations might be just below significance, which would call into question the specificity of their results. I'm not at all trying to suggest that this is the case but I would guess that the readers would like to judge for themselves how convincing the data are and it is thus essential to show these correlations as well, especially since spinal fMRI is still regarded with some scepticism. I am aware that the authors supplied the data that underlie*
Figure 4*, but I couldn't analyse them, as none of my browsers allowed a correct download of the .mat file*.*”*

We agree with this point, and the revised figure no longer zeros out non-significant correlations.

*But more importantly*:

*(b) “Almost all the correlations that are expected to be zero include the white matter. It is frankly ridiculous to regress out the white matter signal in pre-processing, and then say that there is no functional correlation within white matter. This is surely guaranteed to be the case. Without the white matter squares though, this figure would be simply two numbers, which wouldn't make much of a figure*.*”*

White matter regression is important to suppress global signal changes that would otherwise obscure focal correlations, but it is also typically regarded as a somewhat controversial preprocessing step because it can alter connectivity patterns in the brain.

For this latter reason we felt that it is important to expand upon the topic of regressions via analyses of data processed with 11 additional permutations of preprocessing steps (presented as supplementary figures). These additional analyses revealed that there are in fact no significant temporal correlations between white and gray matter sub-regions before CSF regression, and white matter regression does not significantly affect the correlations between gray and white matter. However, we discovered that white matter regression does affect correlations within gray matter and within white matter, which we explain in the supplementary figure legends.

*The second set of important comments pertains to the group analysis in*
Figure 4.

*“The statistical tests in b are not sensible. All data are averaged across slices and subjects, artificially increasing degrees of freedom, reducing standard errors and leading to completely unreasonable p-values. This is clearly unreasonable. The correct analysis averages across slices within subject, and then performs the statistical test across subjects like any normal ROI-based analysis. There are several reasons why this analysis is not reasonable, but the most obvious is that the slices are not independent. Others include the fact that even if the slices were independent and heteroschedastically distributed, the nature of the random effect that would be inferred is not the population of individuals*, *but rather the population of slices”*

*And*:

*“Despite the authors' arguments in their response, I still argue that it is not valid at the group level to test across slices and subjects. They should average across slices so that they test a set of 19 data-points (corresponding to the number of subjects) against zero, and not 228 (corresponding to the number of subjects times the number of slices), as they still do*.*”*

*After discussion, we would like to see the traditional analysis. If possible we would like to see the results of the simplest possible analysis which is a t-test on the first-level z-scores, as well as the non-parametric analysis proposed (but with the new slice averages, rather than the original slices)*.

We agree with these comments and our new analyses average across slices for each subject before testing for statistical significance against zero. After discussions with the reviewing editor on the challenges of group-level analyses in the spinal cord, we have prepared a new Figure 4 to present examples of representative single-voxel correlations across subjects.

*3) We think it very odd that*
Figure 4
*does not contain any spatial or spectral evidence at the group level*.

*(a) If possible, we would therefore like to see the group average spatial maps of the correlations for seed points in the 2 dorsal and 2 ventral horns. Again, the average should be performed within subject first, and then significance should be determined across subjects*.

*(b) We would like to see the power spectra for the time-series of the “resting functional connectivity”. We believe this is important information, given what is known about these spectra for cortical resting activity*.

We agree with these comments and now include power spectra for gray and white matter sub-regions as figure supplements 2, 4, and 6 for Figure 5 (previously Figure 4).

Power spectra were naturally variable for individual sub-regions at a single-subject level because noise power reflects fluctuations from both extraneous sources (lower TSNR) as well as genuine low-frequency connectivity. Furthermore, unambiguously interpreting the spectrum of an individual time series is simply not possible because power from noise sources is aliased. Therefore, after much consideration we felt that it would be most meaningful to focus on the median group-level findings, which are interpretable to a degree even though the spectra are aliased. These results are indeed interesting and further strengthen this manuscript.

*Two further comments from review*:

*4) I am still confused as to how the seed voxel for each individual subject analysis in each gray matter horn was chosen. The authors nicely describe how this was done for the group analysis, but still give no information for the individual analyses*.

The individual subject analyses also used these eroded sub-regions. This has been clarified in the manuscript.

*5) The authors' new choice of threshold for the group analysis (previously: p < 0.01, now: p < 0.000001) seems to be a bit arbitrary. They could just use p < 0.05 with Bonferroni or FDR correction for the number of tests (28 in this case). In my eyes, this would be a more appropriate and well-motivated threshold*.

The group analyses now identify significance at a threshold of p < 0.05 or p < 0.01 with Bonferroni correction (# of tests = 28).

[Editors’ note: the author responses to the previous round of peer review follow.]

We thank the editors and reviewers for their constructive comments and suggestions that have improved this manuscript significantly. We are pleased that this work is viewed as potentially of high interest, that a referee confirms that this is the “first proper resting-state fMRI study of the spinal cord”, and that the editors state that eLife is in principle interested in publishing this work. We appreciate the opportunity to address the weaknesses of the original manuscript and submit a substantially revised version.

The decision letter highlights three major weaknesses as 1) the small size of the cohort, 2) the need for additional analyses, and 3) doubts about the statistical robustness of some of the results. The third weakness is also in part related to the first weakness. We have addressed these weaknesses as follows:

• *Cohort size*. Since the original submission in January, we have acquired resting state spinal fMRI data from a dozen more subjects. We processed these additional 12 data sets and combined the results with the previous 7 data sets, for an updated cohort of 19 healthy volunteers. If a sample size of approximately 20 should result in asymptotically reproducible results, then we believe that our new cohort of 19 subjects should provide robust results of functional connectivity.

• *Additional analyses*. Additional analyses were performed to measure temporal signal-to-noise ratio (TSNR) at two points along the processing pipeline: before and after the application of white matter and CSF “regressors of no interest”. We observed a 32% increase in mean TSNR between these two steps, demonstrating the importance of such regressors to characterize and remove structured noise sources.

• *Statistical robustness of results*. The threshold for statistical significance in the group analysis was decreased significantly (from p < 0.01 to p < 10^-6^), and with this threshold our findings of gray matter connectivity between ventral horns and between dorsal horns are essentially unchanged. Furthermore, the observation that we obtain nearly identical results with a subset of 7 subjects (from the original group analysis) as with all 19 subjects (from the revised group analysis) adds an additional degree of confidence because this shows that our findings are relatively consistent across subjects and not driven by only one or a few subjects.